# The effect of the sugar metabolism on *Leishmania infantum* promastigotes inside the gut of *Lutzomyia longipalpis*: A sweet relationship?

**Sarah Hendrickx**◉*, **Guy Caljon**◉*

Laboratory of Microbiology, Parasitology and Hygiene (LMPH), University of Antwerp, Antwerp, Belgium

* Sarah.Hendrickx@uantwerpen.be (SH); Guy.Caljon@uantwerpen.be (GC)

## Abstract

It is well-known that *Leishmania* parasites can alter the behavior of the sand fly vector in order to increase their transmission potential. However, little is known about the contribution of the infecting host's blood composition on subsequent sand fly infection and survival. This study focused on the host's glucose metabolism and the insulin/insulin-like growth factor 1 (IGF-1) pathway as both metabolic processes are known to impact vector-parasite interactions of other protozoa and insect species. The focus of this study was inspired by the observation that the glycemic levels in the blood of infected Syrian golden hamsters inversely correlated to splenic and hepatic parasite burdens. To evaluate the biological impact of these findings on further transmission, *Lutzomyia longipalpis* sand flies were infected with blood that was artificially supplemented with different physiological concentrations of several monosaccharides, insulin or IGF-1. Normoglycemic levels resulted in transiently higher parasite loads and faster appearance of metacyclics, whereas higher carbohydrate and insulin/IGF-1 levels favored sand fly survival. Although the recorded effects were modest or transient of nature, these observations support the concept that the host blood biochemistry may affect *Leishmania* transmission and sand fly longevity.

## Author summary

Past research on the interaction between the *Leishmania* parasite and the sand fly vector has revealed that *Leishmania* is capable of changing vector behavior to favor transmission of parasites in the environment. Little is known about the impact of host blood composition on parasite development inside the vector and on vector survival. Here, we showed that parasite burdens in the spleen and the liver inversely correlated to the serum blood glucose levels of infected animals, which triggered us to further investigate the effect of blood monosaccharides, insulin and insulin-like growth factor 1 (IGF-1) on sand fly infection and survival. We demonstrated that normal serum glucose levels in the initial parasitized blood meal resulted in transiently higher parasite loads and a faster appearance of infectious parasites, whereas higher sugar and insulin/IGF-1 levels favored sand fly

**Data Availability Statement:** All relevant data are within the manuscript and its Supporting Information files.

**Funding:** This work was funded by research funds of the University of Antwerp (https://www.uantwerpen.be/en/) TT-ZAPBOF 33049 (GC), TOP-BOF 35017 (GC) and KP-BOF 41544 (SH). The funders had no role in study design, data collection and analysis, decision to publish, or preparation of the manuscript.

**Competing interests:** The authors have declared that no competing interests exist.

survival, which supports the concept that the host blood biochemistry may affect *Leishmania* transmission and sand fly longevity.

## Introduction

Leishmaniasis is a neglected tropical disease caused by the protozoon *Leishmania* parasite. Depending on the infecting parasite species the disease manifests itself as cutaneous leishmaniasis (causing skin lesions), mucocutaneous leishmaniasis (destroying cartilage and soft tissue of the nose, pharynx and palate) or visceral leishmaniasis (affecting spleen, liver and bone marrow). This family of related diseases is prevalent in over 98 countries and, although a lot of infections remain asymptomatic, the visceral disease still causes ten thousands of deaths each year. Disease control mainly relies on prompt and accurate diagnosis and treatment and, as the parasite is transmitted through the bites of female sand flies of the genera *Phlebotomus* and *Lutzomyia*, on vector control. Infection with *Leishmania* parasites is known to provoke several changes in the fly's feeding behaviour, hereby increasing the insect's biting persistence and its tendency to feed on multiple hosts, thus boosting parasite transmission [1,2].

For *Plasmodium*, a link has been demonstrated between the presence of human hormones, cytokines, and growth factors that are ingested with the parasitized blood meal and the infection dynamics in the mosquito and transmission potential to other hosts [3]. First of all, low glucose levels are associated with impaired parasite growth and proliferation *in vitro* [4]. Secondly, mice having type 2 diabetes, which is associated with elevated steady-state blood glucose levels, show an increased transmission potential to mosquitos although no significant differences in circulating gametocyte levels were noted [5]. The higher glucose levels have shown to increase the attractiveness of the host for vector feeding [6] via the production of several mosquito-attracting volatiles [7,8]. Thirdly, *Plasmodium* infections are often associated with hypoglycaemia and hyperinsulinemia. By altering the insulin/insulin-like growth factor (IGF) signalling (IIS) pathway inside the vector gut, human insulin levels not only increase mosquito susceptibility to *Plasmodium* infection, but also alter vector immunity favouring mosquito survival and parasite transmission potential [9,10].

This highly conserved IIS pathway, which is known to be implicated in *Plasmodium*-mosquito interactions, also regulates the metabolism, development, lifespan and immunity in other organisms, such as *Drosophila*. Although homologs of the IIS pathway were also identified in sand flies [11], little research hitherto has focused on *Lutzomyia* and *Phlebotomus spp*., leaving a gap in our understanding of the sand fly immune response to *Leishmania* infection and the effects of the blood composition on subsequent parasite infection and transmission.

Nevertheless, previous research also showed a positive correlation between the glucose availability in the culture medium and *in vitro* promastigote growth for *Leishmania* [12]. In addition, decreased glucose concentrations because of malnourishment are well known to decrease immune responses during active leishmaniasis and therefore increase host susceptibility to infection both in experimental animal models and in humans [13–17]. Whether infection can also influence glucose levels in the host and whether prandial glucose levels might impact the infection in the sand fly vector have not been investigated so far. This study therefore aimed to characterize the potential relationship between the host's glucose/insulin metabolism and its effects on *Leishmania* infection in the sand fly vector and on subsequent parasite transmission.

## Materials and methods

### Ethics statement

The use of laboratory rodents was carried out in strict accordance to all mandatory guidelines (EU directives, including the Revised Directive 2010/63/EU on the Protection of Animals used for Scientific Purposes that came into force on 01/01/2013, and the declaration of Helsinki in its latest version) and was approved by the ethical committee of the University of Antwerp, Belgium (UA-ECD 2020–17 (05-05-2020).

### Animals, sand flies and parasites

Female Swiss mice (BW 20–25 g) and Syrian golden hamsters (100–120 g) of six-eight weeks old were purchased from Janvier (France) and kept in quarantine for at least 5 days before infection or blood collection. Food for laboratory rodents and drinking water were available *ad libitum*.

A *Lutzomyia longipalpis* sand fly colony was initiated with the kind help of NIH-NIAID (Prof. Shaden Kamhawi and Prof. Jesus Valenzuela) and maintained at the University of Antwerp under standard conditions (26˚C, >75% humidity, in the dark) with provision of a 30% glucose solution *ad libitum* [18]. For infection and transmission experiments, 3- to 5-day old females from generations 21 to 34 were used.

A recent clinical *Leishmania infantum* isolate (MHOM/ES/2016/LLM-2346) was kindly provided by Dr. J. Moreno from the World Health Organization Collaborate Centre (WHOCC, Madrid, Spain). Promastigotes were routinely cultured in T25 culture flasks containing 5 mL of HOMEM hemoflagellate medium, supplemented with 10% inactivated fetal bovine serum (iFBS), and were subcultured twice weekly. As parasite virulence is known to rapidly decline upon prolonged *in vitro* passage, infections were caried out using low-passage promastigotes (<5 passages). To obtain highly virulent *ex vivo* amastigotes, parasites were also expanded *in vivo* in hamsters. For this purpose, hamsters were infected intracardially with an infection inoculum containing $2 \times 10^7$ parasites/100 μL RPMI-1640. Six to 8 weeks post infection, *ex vivo* amastigotes could be harvested from the spleen, quantified using the method described by Stauber [19] and suspended at a final concentrations of $2 \times 10^7$ amastigotes/100μL in RPMI-1640 to initiate novel infections.

### Chemicals

Human recombinant IGF-1 was purchased from Thermofisher scientific (Waltham, Massachusetts, US), while recombinant human insulin, glucose, fructose and galactose were purchased from Sigma-Aldrich (Saint Louis, Missouri, US). Culture media (HOMEM, RPMI-1640, phosphate buffered saline (PBS), penicillin-streptomycin solution, L-glutamine and Albumax II) were purchased from Life Technologies.

### Hamster studies

To evaluate whether parasite burdens in the main target organs (spleen, liver and bone marrow) could be correlated to serum glucose levels, groups of three hamsters were infected intracardially with a 1:10 dilution series of *ex vivo* amastigotes in RPMI-1640 ranging from $10^8$ to $10^4$ parasites/hamster. At 6 weeks post infection (appearance of clinical symptoms in the group infected with the highest infection dose) sublingual blood was collected to determine serum glucose levels. Hamsters were then euthanized with $CO_2$ and spleen, liver and bone marrow were harvested and used to prepare tissue imprints on glass microscopic slides. The slides were fixed with methanol and stained with a 1:5 Giemsa dilution in water, allowing the

microscopic determination of parasite tissue burdens [expressed as Leishman Donovan Units for liver and spleen (LDU) and as the average number of amastigotes per macrophage for bone marrow].

## Promastigote growth curves

To generate promastigote growth curves, promastigotes were counted using a KOVA counting chamber and seeded in new T25 flasks containing exactly 5 mL promastigote medium at a density of $5 \times 10^5$ promastigotes/mL. To evaluate the growth effects caused by the different types of monosaccharides, IGF-1 and insulin, HOMEM medium without glucose or iFBS, but supplemented with 5 g /L Albumax II, was used. This medium was then spiked with the desired concentrations of glucose, [70 mg/dL (low) or 180 mg/dL (high)], fructose [0.005 nM (low) or 0.15 nM (high)], galactose [10 mg/dL (low) or 100 mg/dL(high)], IGF-1 (50 ng/mL) or insulin (100 mIU/L). Cultures seeded in HOMEM containing Albumax II without the additives were included as controls. Cultures were incubated at 25˚C for 10 days. Each 24h, the parasite density in the cultures was counted using a KOVA counting chamber.

## Preparation of blood

Blood was collected via intracardial puncture of euthanized, fasted Swiss mice using heparin-coated syringes (Becton Dickinson). Upon its collection, blood was kept on ice. It was first centrifuged at 4˚C at 930×$g$ for 10 minutes to separate the cells from the plasma. The cells were washed twice with ice-cold PBS and again centrifuged at 4˚C at 930×$g$ for 10 minutes. The collected plasma was placed at 56˚C for 45–60 minutes to allow inactivation of complement. Afterwards, the plasma was chilled on ice and again mixed with the cells. Finally, 30 μL penicillin/streptomycin was added per mL of blood. Blood glucose levels were evaluated using the OneTouch Verio glucose meter and blood was only used if glucose levels were > 70 mg/dL.

For the experiments with infected blood, 24h-old log phase promastigotes were centrifuged and washed with PBS at 2850×$g$ for 15 minutes at room temperature. After the last washing step, the PBS was removed and the promastigotes were quantified using a KOVA-counting chamber. The blood was infected by adding $5 \times 10^6$ promastigotes/mL.

## Spiking of blood

The blood was then spiked with the different proteins and sugars at a physiological range between a minimum (low), a medium (med) and a maximum (high) concentration (Table 1). The medium concentration is chosen to correspond to the normal physiological concentration of a healthy human individual.

## Sand fly infections

Sand flies were deprived of sugar 14–16 h before infection. For artificial infections, 200–300 flies were placed in infection cups before infection. These small, plastic cups were covered with

**Table 1. Range of physiological concentrations used from the different monosaccharides and proteins.**

|  | Min | Med | Max | References |
|---|---|---|---|---|
| Glucose (mg/dL) | 70 | 125 | 180 | [20] |
| Fructose (mM) | 0.005 | 0.03 | 0.15 | [21] |
| Galactose (mg/dL) | 10 | 50 | 100 | [22] |
| Insulin (mIU/L) | 25 | 100 | 230 | [23–26] |
| IGF-1 (μM) | 0.013 | 0.06 | 0.13 | [3,9] |

an insect mesh through which flies were offered access to sugar or blood meals. The prepared blood was placed into a glass feeder device covered with a chick skin-membrane and placed on the mesh of the infection cup for 2 h, allowing the flies to take a blood meal. Twenty-four hours after infection blood-fed females were isolated and further maintained at 26˚C to evaluate the course of infection over time. To avoid overcrowding and associated premature death, flies were kept with a maximum of 150 females in one infection cup. During the follow-up period of the experiment flies were given access to a 30% sucrose solution *ad libitum*, to mimic their access to sugary plant secretions in nature.

### Evaluation sand fly survival

Sand fly survival was evaluated at days 2, 5, 7 and 9 after infection by counting the number of death flies per condition. Based on the number of dead flies removed from the cages, Kaplan-Meier curves were constructed using Graphpad prism version 6.0.

### Evaluation parasite load and metacyclogenesis in sand fly

To evaluate the total parasite load in the fly gut and the rate of metacyclogenesis, sand fly guts were dissected at days 2, 5, 7 and 9 after infection. The gut was macerated in 50 µL PBS with a plastic pestle and the number of parasites inside the gut was microscopically determined using a KOVA counting chamber. In parallel, the percentage of metacyclics was determined microscopically in the same KOVA counting chamber based on parasite morphology and motility as established elsewhere [27,28]. All experimental infections were performed with an independent repeat for each condition. To avoid bias during counting and microscopic evaluation of morphology and motility, groups were randomized by an independent researcher until data analysis.

### Statistical analysis

All statistical analyses were performed using SPSS and Graphpad Prism version 6.00 software. Statistical differences between the different experimental sand fly groups were determined by 2-way ANOVA, whereas differences between groups in the Kaplan-Meier survival analyses were determined by Pearson Chi-Square analysis. The potential correlation between serum glucose levels and parasite tissue burdens and transmission were analyzed using a Spearman ranked test. Tests were considered statistically significant if $p < 0.05$.

## Results

### Host blood glucose levels are correlated to parasite burdens in liver and spleen

When comparing the blood glucose levels of the host with the respective parasite burdens recorded in the main target organs (spleen, liver and bone marrow) a moderate correlation ($0.4 < |r| < 0.6$) could be found for the liver (r = -0.5573; p = 0.0265 (*)) and the spleen (r = -0.5714; p = 0.02286 (*)) (Fig 1A and 1B), which was enhanced when evaluating the number of amastigotes/nucleus [liver (r = -0.5914; p = 0.0227 (*)); spleen (r = -0.7; p = 0.0048 (**))] (Fig 1C and 1D). This correlation could not be corroborated in the bone marrow (r = -0.2806; p = 0.3087) (Fig 1E)

### Monosaccharide levels do not significantly impact promastigote growth in vitro

Despite earlier reports indicating a possible growth-stimulating effect of glucose on *in vitro* promastigote cultures [12], no growth-enhancing effect was observed under our experimental conditions upon increasing the monosaccharide concentration in the culture medium

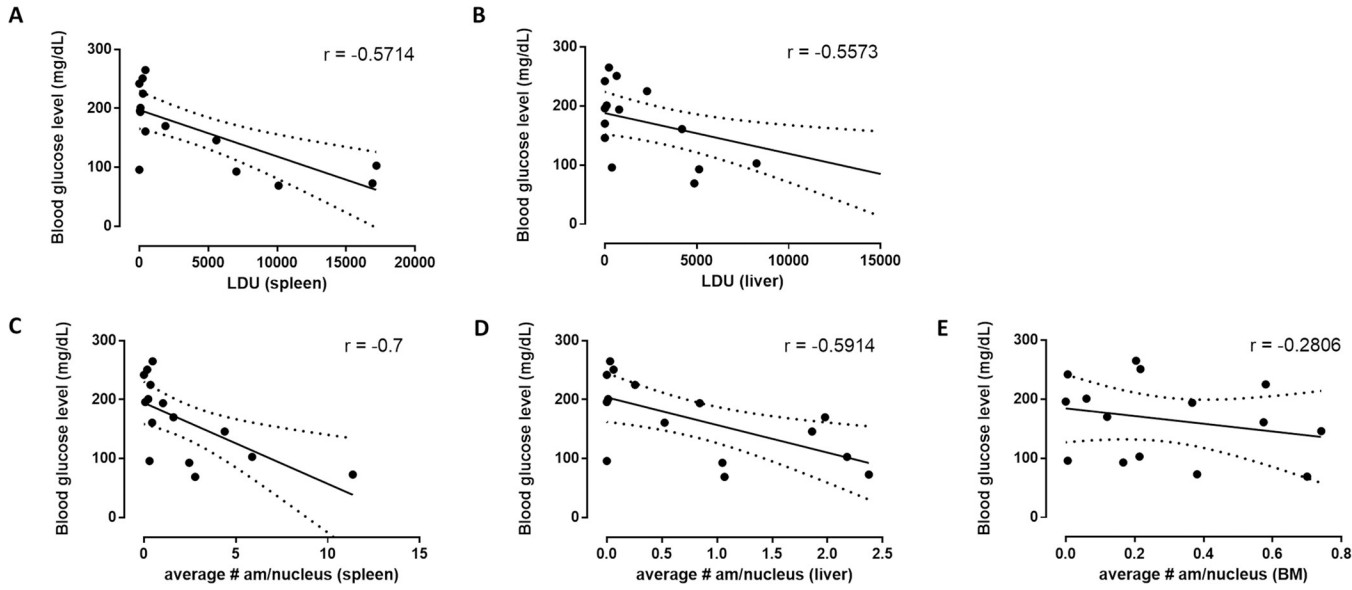

**Fig 1. Blood glucoses levels in the blood of infected hamsters in relation to the parasite burdens in spleen (A-C), liver (B-D) and bone marrow (E).** Six weeks after infection sublingual blood was collected from hamsters infected with different parasite burdens and the blood glucose concentration was determined. Hamsters were autopsied afterwards and the parasite burdens in the target main organs (liver, spleen and bone marrow) were determined based on microscopic counting of the number of amastigotes on Giemsa-stained tissue imprints. Results were analysed using the nonparametric Spearman correlation analysis (correlation coefficients r indicated in the graphs) and with inclusion of a linear regression. A weak to moderate correlation between both parameters could be found in the spleen and the liver. For the bone marrow both parameters could not be correlated. Results are based on data derived from 3 biological repeats in 15 individual hamsters (5 groups of 3 animals). Each group received a different infection inoculum containing either $10^8$, $10^7$, $10^6$, $10^5$ or $10^4$ parasites.

(S1 Fig). Similarly, the addition of IGF-1 or insulin to the *in vitro* culture medium also had no significant effect on promastigote growth.

## Normal blood sugar levels are most compatible with Leishmania development in the sand fly

When evaluating the effect of the various monosaccharide concentrations on infection inside the vector gut over time (Fig 2), little effects were observed. On day 5 and day 7 of the

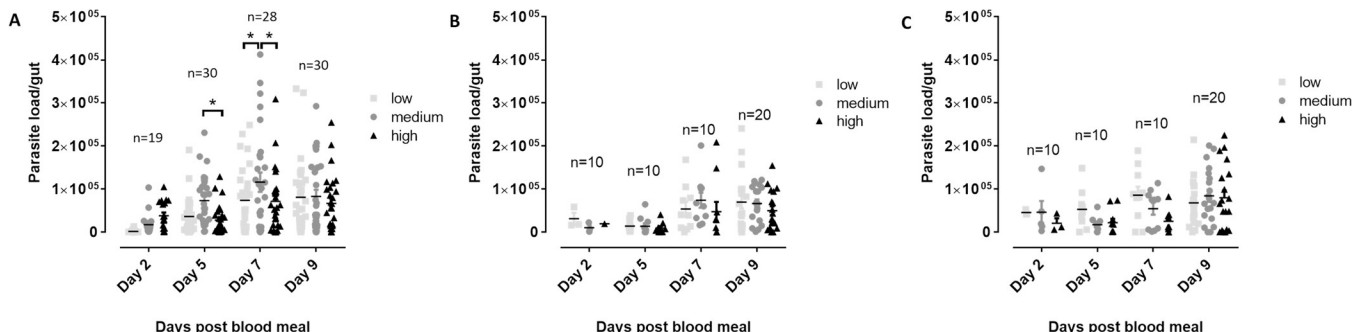

**Fig 2. Effect of different concentrations of glucose (A), fructose (B) and galactose (C) in the infective blood meal on the *Leishmania* parasite load in the sand fly gut during infection.** Sand flies were are allowed to feed on infected blood spiked with different saccharide concentrations within the physiological range (low, medium, high). The infection was followed up over time by dissecting a fixed number of flies at different time points after infection (days 2, 5, 7 and 9) and microscopic counting of the number of parasites. Medium glucose concentrations significantly increase the parasite load in the vector gut at days 5 and 7 post infection. Depicted parasite loads are the result of at least two independent experiments and are expressed as average parasite load/gut ± standard error of the mean. The number of flies dissected at each time point for each saccharide is specified in the respective graphs (*: $p < 0.05$).

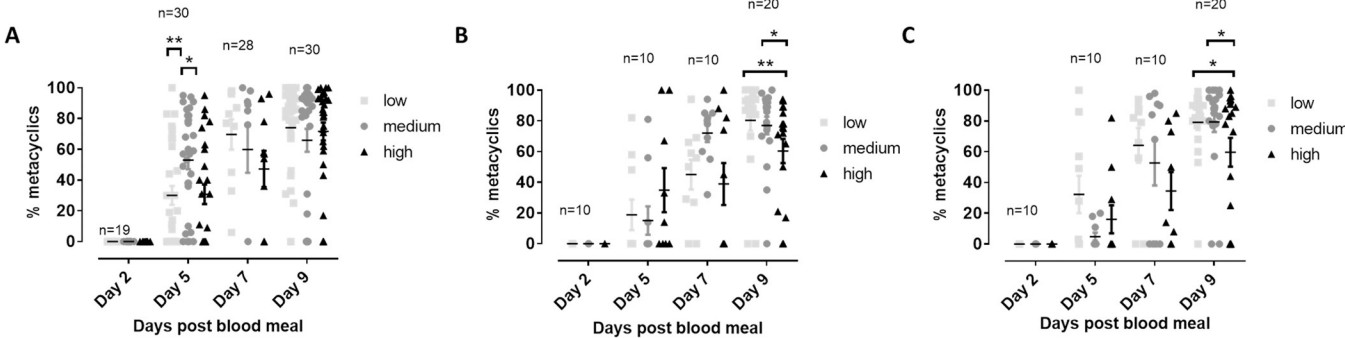

**Fig 3. Effect of different concentrations of glucose (A), fructose (B) and galactose(C) in the infective blood meal on the percentage of metacyclic** *Leishmania* **parasites in the sand fly gut during infection.** Sand flies were are allowed to feed on infected blood spiked with different saccharide concentrations within the physiological range (low, medium, high). The infection was followed up over time by dissecting a fixed number of flies at different time points after infection (days 2, 5, 7 and 9) and microscopic evaluation of the number of metacyclic parasites. Significant differences can be observed at day 5, where medium glucose levels lead to a higher percentage of metacyclics. At day 9, high concentrations of fructose and galactose in the blood are responsible for a reduced metacyclogenesis. Percentages are the result of at least two independent experiments and are expressed as average percentage metacyclics ± standard error of the mean. The number of flies dissected at each time point for each saccharide is specified in the respective graphs (*: p < 0.05; **: p < 0.01).

infection, medium concentrations of glucose, which correspond to the normal physiological concentrations found in healthy human beings, resulted in a slightly higher parasite load in comparison to the hypo- and hyperglycemic conditions.

Despite the limited effects of monosaccharide concentrations on gut parasite load, the effect on metacyclogenesis was found more pronounced (Fig 3). Medium concentrations of glucose led to the earlier development of metacyclics in the vector gut, which suggests again that normal physiological glucose concentrations are optimal for transmission. For fructose and galactose, high saccharide concentrations exerted a negative impact on metacyclogenesis.

## Increased blood sugar levels enhance sand fly survival

To evaluate whether the addition of glucose to the blood meal also had an impact on sand fly survival, the number of dead sand flies was recorded during the infection experiments and a Kaplan-Meier analysis was performed (Fig 4). This analysis revealed a significant benefit of feeding on the high glucose concentrations for sand fly survival. Flies fed on hypoglycemic blood had the lowest survival rates.

## Blood insulin and IGF1 levels do not impact Leishmania development in the sand fly

Spiking the infective blood meal with different physiological concentrations of insulin did not impact parasite load or metacyclogenesis (Fig 5A and 5B). Changing IGF-1 concentrations in the infective blood meal also did not lead to differences in gut establishment and parasite maturation (Fig 5C and 5D). Only on day 7 post infection, parasite levels and percentage metacyclics tend to be highest in the flies fed on blood with normal IGF-1 concentrations (medium). Higher IGF-1 levels seem to rather negatively impact the infection.

## High insulin or IGF-1 levels favor sand fly survival

Although the blood insulin concentration has no effect on parasite infection in the sand fly gut (Fig 5A and 5B), medium and high insulin concentrations in the blood meal were able to significantly enhance sand fly survival over the course of the experiment (Fig 6A). When

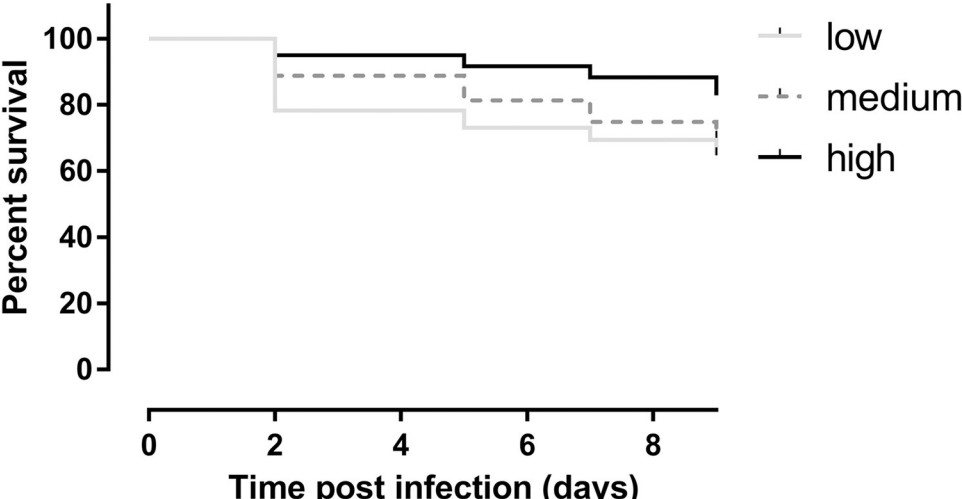

**Fig 4. Kaplan-Meier survival curve comparing survival of sand flies that were infected with blood spiked with different concentrations of glucose.** A significant difference could be observed between the different groups. Addition of high concentrations of glucose in the infective blood meal significantly enhances fly survival over the course of infection compared to medium (***: $p < 0.005$) and low concentrations (****: $p < 0.001$). This analysis is the result of three independent experiments (n = 193 low, n = 294 medium, and n = 239 high).

assessing sand fly survival upon blood feeding with different IGF-1 concentrations over time (Fig 6B), IGF-1 initially does not seem to impact vector survival. However, final survival was highest in the group feeding blood with high IGF-1 concentrations (day 9).

## Discussion

With the increasing spread of Western diet and lifestyle worldwide, overweight and the so-called "civilization diseases", such as diabetes mellitus type 2, are also rapidly increasing in several developing countries [29]. The emerging co-occurrence of type 2 diabetes mellitus and tropical infectious diseases, however, may have substantial implications as type 2 diabetes mellitus is known to increase the body's susceptibility to common infections [30]. Infections also significantly impact host immunological and physiological parameters. In this study, artificial infection of Syrian golden hamsters with *L. infantum* parasites revealed a moderate inverse correlation between serum blood glucose levels and tissue burdens in the spleen and liver. This finding prompted us to further evaluate the impact of the host serum composition on sand fly infection and survival. For other infectious diseases, such as malaria, the presence of human hormones, cytokines, and growth factors (such as glucose and insulin) in the infectious blood meal can indeed alter the course of infection, hereby affecting vector physiology and parasite transmission [3].

Carbohydrates are not only very important for sand fly biology, but also for *Leishmania* development and transmission. Although female sand flies need a blood meal to initiate ovarian development and oviposition, sand flies mainly feed on plant juices and sugary secretions. These ingested sugars are very important for subsequent parasite development and anterior migration in the sand fly gut [31–37] and are key drivers for metacyclogenesis [12] and the formation of the parasite's LPG coat [37–41], which is altered upon parasite detachment from the fly's midgut [41–45]. Moreover, sand fly survival decreases in the absence of sugars [31].

As saccharides may clearly impact parasite development inside the sand fly vector, their impact on promastigote multiplication was first evaluated *in vitro*. Under the used

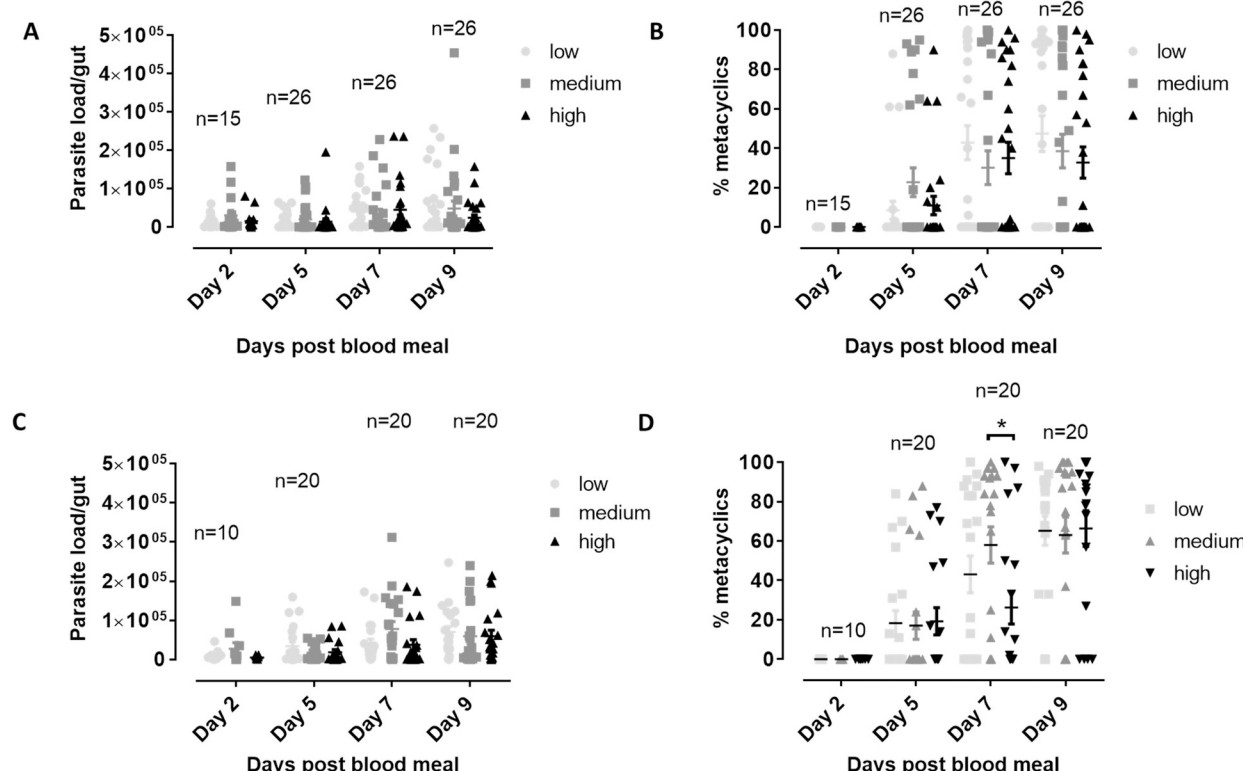

**Fig 5. The effect of different insulin and IGF-1 concentrations in the infective blood meal on the parasite load (A) and the percentage of metacyclic parasites (B) during infection.** Sand flies were are allowed to feed on infected blood spiked with different concentrations on insulin or IGF-1 within the physiological range (low, medium, high). The infection was followed up over time by dissecting a fixed number of flies at different time points after infection (days 2, 5, 7 and 9) and microscopic counting of the number of parasites and microscopic evaluation of the number of metacyclic parasites. Depicted parasite loads and percentage metacyclics are the result of at least two independent experiments and are expressed as average ± standard error of the mean. The number of flies dissected at each time point for each saccharide is specified in the respective graphs (*: p < 0.05; **: p < 0.01).

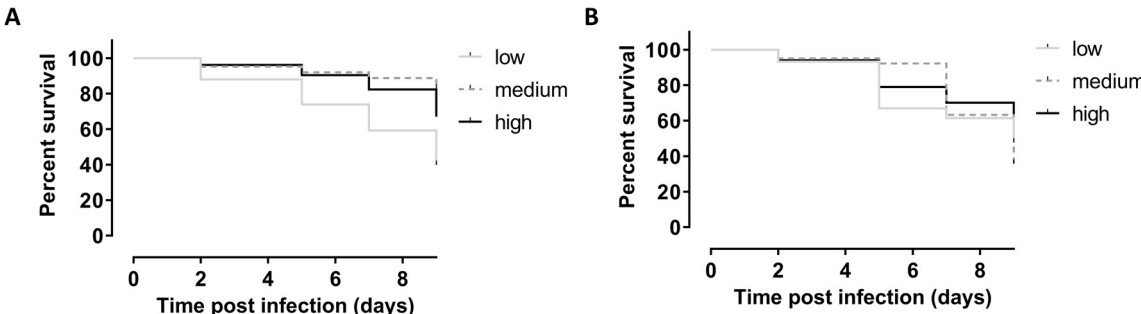

**Fig 6. Kaplan-Meier survival curve comparing sand fly survival of flies that were infected with blood spiked with different concentrations of insulin (A) and IGF-1 (B).** A significant difference could be observed between the different groups. Addition of low concentrations of insulin in the infective blood meal significantly decreased fly survival over the course of infection compared to medium (****: p < 0.001) and high concentrations (****: p < 0.001). Addition of high concentrations of IGF-1 in the infective blood meal significantly decreased fly survival over the course of infection compared to low (*: p < 0.05) and medium concentrations (***: p < 0.005). For insulin, this analysis is the result of two independent experiments and is based on 444 (low), 462 (medium), and 477 (high) observations, while for IGF-1, the analysis is the result of two independent experiments (n = 400 low, n = 387 medium, and n = 385 high).

experimental conditions, no effect of the addition of high saccharide concentrations in the culture medium could be observed. When assessing the effect of monosaccharide addition to the infective blood meal, it was therefore not surprising to observe that higher concentrations of monosaccharides did not enhance parasite multiplication. Normal physiological glucose concentrations were compatible with the highest parasite load in the sand fly gut and an increased proportion of metacyclics early in infection (day 5). Our results also show that intake of a sugar-rich blood meal increases the survival of *L. longipalpis*, a phenomenon that has already been reported for Drosophila [46]. Dietary glucose additionally enhanced immunity against enteric infections with the Gram-negative *Vibrio cholerae* bacteria in adult *Drosophila* [47]. Whether this effect in *L. longipalpis* is also caused by an enhancing effect on insect immunity, remains to be investigated.

Given the differences in the biology and ecology each pathogen-vector interaction, research on phlebotomines is essential to identify parallels or unique features of the *Leishmania* infection and transmission process. Hosts infected with *Plasmodium* demonstrate increased blood insulin concentrations, which trigger the insulin/IGF-1 signaling pathway in *Anopheles* mosquitos, hereby increasing mosquito susceptibility to infection, lifespan and transmission [10,48]. For *Leishmania*, increased insulin concentrations did not result in increased vector susceptibility to infection. However, high and medium concentrations of insulin seem to extend the lifespan of the vector similar to what was observed for malaria mosquitos [10]. IGF-1 on the other hand, was shown to render mosquitos more resistant against *Plasmodium falciparum* infections and to increase the mosquito lifespan [3,9]. In line with these findings, high IGF-1 concentrations during infection of *L. longipalpis* reduced parasite loads and metacyclogenesis at day 7 post infection and an enhanced sand fly survival by the end of the experiment. Highest parasite levels and proportion of metacyclics were observed with normal IGF-1 levels. Although our results corroborate earlier work performed for *Plasmodium*, the effect of this specific factor on *L. infantum* infection in *L. longipalpis* is considerably less pronounced. Possibly, the differences observed between *L. longipalpis* sand flies, *Drosophila* fruit flies and *Anopheles* mosquitos are caused by differences in their immune systems. For *Drosophila* and *Anopheles* most immunological pathways involved in symbiosis with gut microbiota and defense against parasitic infections are fairly well characterized [49–52], with the Toll pathway and immune deficiency (Imd) pathway being important contributors to *Drosophila* and mosquito immunity. In *Drosophilla*, recent research even revealed the implication of Imd pathway activation in metabolic regulation (hyperglycemia, depleted fat reserves, reduced survival, reduced size and developmental delays) and immunity (increased susceptibility to infections) [53–58]. Past research on two *L. longipalpis*-derived cell lines, LL-5 and Lulo, had already been suggestive for the potential involvement of both these pathways in innate immune responsiveness against *Leishmania* [59]. Recent research indeed corroborated the upregalation of both pathways in sand flies upon a blood meal, they were not significantly upregulated in the presence of *Leishmania* and most likely react to the increase in sand fly midgut bacteria in the presence of blood [60–62]. However, the link between parasite survival, gut microbiota and the immune responses mounted by the sand fly definitely needs further investigation.

Collectively, the observations in this study show that host blood biochemical parameters of the glucose metabolism impact sand fly longevity and cause modest, transient effects in *Leishmania* development in the insect gut. In general, higher physiological carbohydrate and insulin/IGF-1 seem to enhance fly survival whereas normoglycemic conditions transiently favor early development of metacyclics. Although these observations may have a limited epidemiological impact, they support the concept that host blood biochemical parameters may affect *Leishmania* transmission and sand fly longevity.

## Supporting information

**S1 Fig. Comparison of *in vitro* promastigote growth of *L. infantum* cultured in either medium without additives (ctr) or in media containing low of high levels of glucose (A), fructose (B) or galactose (C) or insulin and IGF-1 (D).** No significant differences in *L. infantum* promastigote growth could be observed. The parasite density at each time point of cultivation is the average of at least two independent experiments run in duplicate ± the standard error of the mean.
(TIF)

## Acknowledgments

The authors wish to thank lab technicians Mathias Sempels for the maintenance of the sand fly colony, Pim-Bart Feijens for the technical assistance with the hamster infections and Mandy Vermont for the technical assistance during the randomization process. LMPH is a partner of the Excellence Centre 'Infla-Med' (www.uantwerpen.be/infla-med).

## Author Contributions

**Conceptualization:** Sarah Hendrickx, Guy Caljon.

**Data curation:** Sarah Hendrickx.

**Formal analysis:** Sarah Hendrickx, Guy Caljon.

**Funding acquisition:** Sarah Hendrickx, Guy Caljon.

**Investigation:** Sarah Hendrickx.

**Methodology:** Sarah Hendrickx.

**Project administration:** Sarah Hendrickx, Guy Caljon.

**Resources:** Guy Caljon.

**Supervision:** Sarah Hendrickx, Guy Caljon.

**Visualization:** Sarah Hendrickx.

**Writing – original draft:** Sarah Hendrickx.

**Writing – review & editing:** Sarah Hendrickx, Guy Caljon.

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
