## [Decision Letter · Decision Letter 0]

2 Dec 2021

Dear Dr. Caljon,

Thank you very much for submitting your manuscript "The effect of the sugar metabolism on Leishmania infantum promastigotes inside the gut of Lutzomyia longipalpis: a sweet relationship?" for consideration at PLOS Neglected Tropical Diseases. As with all papers reviewed by the journal, your manuscript was reviewed by members of the editorial board and by several independent reviewers. In light of the reviews (below this email), we would like to invite the resubmission of a significantly-revised version that takes into account the reviewers' comments. 

Dear colleagues,

thank you for submitting your work to PLOS NTD! As you will find, the reviewers took some issues with your methodology and conclusions, and I ask you to comply fully with their recommendations. In addition, my own review found additional points that should be corrected:

1. Figure 1: the figure lacks the panels D and E that are advertised in the text. Moreover, the results shown in the figure (panel B) do not match your description in the text; the R-square values differ. Lastly, the Figure legend is insufficient and must be amended with all pertinent information needed for understanding your experimental set-up, including sample sizes and number of biological repeats.

2. Figure 2 does not convey important information and should be shifted to the supplementary information.

3. For Figures 3ff, state how many repeat experiments were conducted. The figures only display the number of sandflies that were evaluated. Likewise, please state what measures, e.g. double-blind counting, were taken to avoid observational bias since you relied heavily on microscopic evaluations.

3. line 167: "In parallel, an estimated percentage of metacyclics was calculated based on parasite morphology and motility. " Not sufficient. What were the parameters used to distinguish between procyclics and metacyclics and where were those parameters first described and/or evaluated (Citations)?

4. Please make clear, that the majority of the observed effects are transient, in the text, the conclusions and the abstract.

I look forward to your revised manuscript!

Yours

Joachim Clos

We cannot make any decision about publication until we have seen the revised manuscript and your response to the reviewers' comments. Your revised manuscript is also likely to be sent to reviewers for further evaluation.

Sincerely,

Joachim Clos

Associate Editor

Shan Lv

Deputy Editor

Dear colleagues,

thank you for submitting your work to PLOS NTD! As you will find, the reviewers took some issues with your methodology and conclusions, and I ask you to comply fully with their recommendations. In addition, my own review found additional points that should be corrected:

1. Figure 1: the figure lacks the panels D and E that are advertised in the text. Moreover, the results shown in the figure (panel B) do not match your description in the text; the R-square values differ. Lastly, the Figure legend is insufficient and must be amended with all pertinent information needed for understanding your experimental set-up, including sample sizes and number of biological repeats.

2. Figure 2 does not convey important information and should be shifted to the supplementary information.

3. For Figures 3ff, state how many repeat experiments were conducted. The figures only display the number of sandflies that were evaluated. Likewise, please state what measures, e.g. double-blind counting, were taken to avoid observational bias since you relied heavily on microscopic evaluations.

3. line 167: "In parallel, an estimated percentage of metacyclics was calculated based on parasite morphology and motility. " Not sufficient. What were the parameters used to distinguish between procyclics and metacyclics and where were those parameters first described and/or evaluated (Citations)?

4. Please make clear, that the majority of the observed effects are transient, in the text, the conclusions and the abstract.

I look forward to your revised manuscript!

Yours

Joachim Clos

Reviewer's Responses to Questions

**Key Review Criteria Required for Acceptance?**

**Methods**

-Are the objectives of the study clearly articulated with a clear testable hypothesis stated?

-Is the study design appropriate to address the stated objectives?

-Is the population clearly described and appropriate for the hypothesis being tested?

-Is the sample size sufficient to ensure adequate power to address the hypothesis being tested?

-Were correct statistical analysis used to support conclusions?

-Are there concerns about ethical or regulatory requirements being met?

Reviewer #1: The study has a clear hypothesis that was tested using appropriate methods, but some information should be added. 

I miss the information if infected sand flies were further provided with a 30% glucose solution as the colony. This is important as this additional sugar supply may decrease or fully cover between-group differences of glucose intake in the blood. 

Also, number of sand fly females should be given for each group in Figs. 5 and 7 as this parameter is more important than the number of observations.

Reviewer #2: -Are the objectives of the study clearly articulated with a clear testable hypothesis stated?

Yes

-Is the study design appropriate to address the stated objectives?

Yes

-Is the population clearly described and appropriate for the hypothesis being tested?

Yes

-Is the sample size sufficient to ensure adequate power to address the hypothesis being tested?

Yes

-Were correct statistical analysis used to support conclusions?

Partly, there are some mistakes in the figures regarding the analysis of statistical significance

-Are there concerns about ethical or regulatory requirements being met?

No

**Results**

-Does the analysis presented match the analysis plan?

-Are the results clearly and completely presented?

-Are the figures (Tables, Images) of sufficient quality for clarity?

Reviewer #1: The results are clearly described with exception of the first chapter “Correlation of host blood glucose levels with parasite burdens” (lines 180-185). Here, the text including statistical values does not correspond to Figure 1 and its legend. According to the figure, the correlation was found in the spleen only while in the text, correlation is declared also for the liver. This must be clarified as the statement that the correlation was significant also for the liver repeats in many parts of the manuscript. Also, Figure 1 has only three parts (A, B and C) while in the text, A-E parts are listed. In addition, two kinds of evaluation of parasite load are mentioned in the text but only one is described in Materials and Methods.

Reviewer #2: -Does the analysis presented match the analysis plan?

Yes

-Are the results clearly and completely presented?

No, Figure 1 is incomplete (E+D are missing)

-Are the figures (Tables, Images) of sufficient quality for clarity?

I would recommend to improve all figures.

Figure 1 is incomplete

**Conclusions**

-Are the conclusions supported by the data presented?

-Are the limitations of analysis clearly described?

-Do the authors discuss how these data can be helpful to advance our understanding of the topic under study?

-Is public health relevance addressed?

Reviewer #1: Conclusions are not well supported by data and authors overvalue their findings of the effect of sugars on Leishmania development in sand flies: In Fig. 3 the little effect was demonstrated for medium glucose concentration on days 5 and 7 but not on late infection on day 9 (and no effects were demonstrated for fructose and galactose). In Fig. 4 little effect of glucose was demonstrated for metacyclogenesis as slight differences were seen on day 5 but not on days 7 and 9 (and the negative effect was found for other monosaccharides).

Reviewer #2: -Are the conclusions supported by the data presented?

Yes

-Are the limitations of analysis clearly described?

No. It is necessary to make limitations clear, especially the comparing to other insects/parasites/bacteria must be improved

-Do the authors discuss how these data can be helpful to advance our understanding of the topic under study?

Yes, but minor improvements would be helpful

-Is public health relevance addressed?

Too less, I would recommend to improve this.

**Editorial and Data Presentation Modifications?**

Reviewer #1: Abstract:

 Line 26 – “ and because glycemic levels…..this sentence should be separated as it represents the result of this manuscript, not previously published facts like the beginning of the compound sentence does. 

 Line 28 – “and hepatic” should be deleted as hepatic parasite burdens are not inversely correlated to glycemic levels according to data presented in Figure 1.

The last two sentences of the Abstract should be changed to better correspond to the described results: Higher carbohydrate and insulin/IGF-1 levels favored sand fly survival while all the recorded effects on parasite loads and appearance of metacyclics were modest, temporary and not apparent in late infections. Therefore, these observations do not support the concept that host blood biochemistry can directly affect Leishmania transmission. 

Similarly, also Author Summary and the end of the Discussion should be rewritten.

Figures 2, 5 and 7: in the current black and white presentation, lines for low and medium levels are not clearly distinguishable. I suggest using color presentation or patterned lines.

Lines 337-339: Results obtained with L. longipalpis –derived cell lines should not be interpreted as confirmation of the role of toll and lmd pathways in innate responsiveness of the sand fly against Leishmania. Instead, recent studies with L. longipalpis will be more appropriate for such discussion. 

Discussion is relatively long, in several aspects similar to Introduction and not focused to own results.

Reviewer #2: (No Response)

**Summary and General Comments**

Reviewer #1: The topic of the manuscript is actual; the authors focused on the correlation of host glucose metabolism with leishmania development in phlebotomine sand flies and fly survival. The results support the effect of Leishmania infection on glucose level in the host and the effect of blood sugars and insulin/IGF1 levels on sand fly longevity. On the other hand, effects of blood sugars and insulin/IGF1 levels on parasite loads and appearance of metacyclics were none or too modest, but always temporary and, more importantly, they were not apparent in late infections on day 9. 

However, the “negative results” are also important and should be published with a proper interpretation. 

I suggest major revisions and:

- Clarify better the results of experiments on the correlation of host blood glucose levels with parasite burdens

- Change interpretation of data concerned on the correlation of blood biochemistry with parasite development in sand flies (my suggestions for changing the Abstract are written above).

Reviewer #2: Comments for authors

Major comments

The authors present an interesting study about the impact of host sugar metabolism, i.e.

blood composition, for sandflies and Leishmania parasites. The performed experiments are

reasonable and adequately to prove their message.

- The introduction should be more organized. The examples for the influence of

glucose levels on parasites in general switches between different insects and hosts

without mentioning this and the possible differences (ecological behavior etc.).

- The Introduction of Leishmania parasites in the beginning is too short

- Introduction: The comparison of increasing parasite growth/proliferation due to low

glucose levels with increasing transmission due to high glucose levels is not correct.

Please compare either the growth/proliferation on different glucose levels or the

transmission

- Most figures must be improved

- Figure 1 is not complete, 1 D and 1 E are missing

- Discussion: It is important to differentiate between the different insects and

infections (different parasites, bacteria), this is done too generalized. For example,

Line 319-320: reference 43 is about the immune stimulation in Drosophila due to

Vibrio cholerae infections. The immune response of insects to gram-negative bacteria

and parasites can utilizes different pathways. This should be mentioned in the text.

Minor comments

53 > make sure you don´t mix up sand fly and mosquito behavior, add “mosquito” before

“vector” or

51-54 delete this sentence

59 what about the parasite growth and proliferation in these mice?

71 rephrase (For example: “It is known, that there is a correlation…”)

71-72 positive or negative correlation of glucose availability and growth?

72-75 But this is for infections in general or are there any specific information’s for

leishmania infections?

89 add the age of mice

93 explain shortly standard conditions

95 which generation of the colony was used for experiments?

97: change “dr.” “Dr.”

97: add information about the passage

102/104: standardize to xx/100µl or xx/mL for all

113: add 104

to 108

“parasites per hamster”

126-128: standardize units

130: recommendation to write “10 days” instead of “240h”

139: rephrase “these levels were >70mg/dL” to “glucose level were > 70 mg/dL”

153: add number of flies per infection cup

154: cite or explain infection cup

156: delete “and thus infected”

158: Explain “cup”

181: explain moderate correlation

184/185: Figure 1D and 1E are missing!!!

Figure 1: Information in Figure and Text must match

199: change “of” to “or”

Figure 2: change varieties of statement; Line 207 says “seem to favor” while Line 211 says

“significantly increase” for the same group (medium)

Figure 5: delete “****” from the figure, it´s not possible to do statements like this with more

than two groups

237-238: Change “three independent experiments and is based on 193….” To “three

independent experiments (n=193 low, n=294 medium, n=239 high).”

242: add (Fig 6 A + B)

243: add (Fig 6 C + D)

258-259: rephrase

Figure 7: delete stars above A and B

267-268: see comment for 237-238 > change to n=xxx

273: add reference

273-276: rephrase

308-309: rephrase

319-320: Reference 43 is about a gram-negative bacteria, comparison to a parasite infection

is sustainable, but must be mentioned

321: rephrase

457: delete “43.”

468: change “Imd” to “imd”

PLOS authors have the option to publish the peer review history of their article (what does this mean?). If published, this will include your full peer review and any attached files.

Reviewer #1: No

Reviewer #2: No
---

## [Decision Letter · Decision Letter 1]

2 Mar 2022

Dear Dr Caljon,

We are pleased to inform you that your manuscript 'The effect of the sugar metabolism on Leishmania infantum promastigotes inside the gut of Lutzomyia longipalpis: a sweet relationship?' has been provisionally accepted for publication in PLOS Neglected Tropical Diseases.

Best regards,

Joachim Clos

Associate Editor

Shan Lv

Deputy Editor

Dear Colleagues,

thank you for submitting a revised manuscript and for the close attention you paid to the reviewers' suggestions. It is my pleasure to inform you that your manuscript has now been accepted for publication and I offer my felicitations!

Joachim Clos

Reviewer's Responses to Questions

**Key Review Criteria Required for Acceptance?**

**Methods**

-Are the objectives of the study clearly articulated with a clear testable hypothesis stated?

-Is the study design appropriate to address the stated objectives?

-Is the population clearly described and appropriate for the hypothesis being tested?

-Is the sample size sufficient to ensure adequate power to address the hypothesis being tested?

-Were correct statistical analysis used to support conclusions?

-Are there concerns about ethical or regulatory requirements being met?

Reviewer #1: (No Response)

Reviewer #2: all requested changes have been made

**Results**

-Does the analysis presented match the analysis plan?

-Are the results clearly and completely presented?

-Are the figures (Tables, Images) of sufficient quality for clarity?

Reviewer #1: (No Response)

Reviewer #2: All figures are changed as requested

**Conclusions**

-Are the conclusions supported by the data presented?

-Are the limitations of analysis clearly described?

-Do the authors discuss how these data can be helpful to advance our understanding of the topic under study?

-Is public health relevance addressed?

Reviewer #1: (No Response)

Reviewer #2: The conclusions supported by the presented data. The authors improve the part were they discuss similar effects in other insects/parasites/bacteria as requested, even if this could be still focus more on the limitations by this comparisons.

**Editorial and Data Presentation Modifications?**

Reviewer #1: (No Response)

Reviewer #2: (No Response)

**Summary and General Comments**

Reviewer #1: I appreciate the improvement of the manuscript. My suggestions were accomplished and I am delighted to recommend the manuscript for publication. Authors should just correct a minor typing error - the r value for the liver in the text (Line 202) does not correspond to the value in figure 1B (r = -0.5573 vs -0.5773).

Reviewer #2: (No Response)

PLOS authors have the option to publish the peer review history of their article (what does this mean?). If published, this will include your full peer review and any attached files.

Reviewer #1: No

Reviewer #2: **Yes: **Anna Heitmann

---

## [Editor Report · Acceptance letter]

25 Mar 2022

Dear Prof. Caljon,

We are delighted to inform you that your manuscript, "The effect of the sugar metabolism on Leishmania infantum promastigotes inside the gut of Lutzomyia longipalpis: a sweet relationship?," has been formally accepted for publication in PLOS Neglected Tropical Diseases.

Best regards,

Shaden Kamhawi

co-Editor-in-Chief

Paul Brindley

co-Editor-in-Chief
